# SAMO: Semantic-Aware Model Optimization for Source-Free Domain Adaptive Object Detection

## Abstract

Source-Free Adaptive (SFA) object detection aims to adapt a pre-trained detector from a labeled source domain to an unlabeled target domain without access to source data. To address the performance degradation commonly observed in this setting, we introduce a novel Semantic-Aware Model Optimization (SAMO) framework that explicitly models the interplay between semantic perception and model optimization. Specifically, SAMO first performs semantic perception by categorizing target-domain regions into a confidence spectrum (from high to low) based on pseudo-labels generated by a static base detector. Building on this perception, we design an optimization strategy via detector decomposition and disentangled training. The detector is decoupled into shared modules and a specialized expert group, the latter equipped with a semantically aware gating network. Through disentangled training, expert units and shared modules are independently optimized according to their respective confidence spectrum ranges, thereby maintaining discriminative capacity across diverse semantic levels. Meanwhile, the gating network adaptively integrates expert features through learnable attention weights, enabling dynamic discrimination across semantic categories. This framework effectively alleviates the source-domain performance degradation caused by low-confidence regions, while simultaneously achieving significant improvements in the target domain. Extensive experiments on three domain adaptation benchmarks demonstrate the superior generalization ability of SAMO. Our code will be released.

## 1 Introduction

Deep learning-based object detection has seen significant advancements, with many high-performance detectors such as Faster R-CNN(Ren et al., 2017), FCOS(Tian et al., 2019), and Deformable DETR(Zhu et al., 2021) being proposed. However, these detectors are typically trained and evaluated on large-scale datasets. When applied to new scenarios with significant differences in data distribution from the supervised training dataset, the performance of these detectors often deteriorates. Moreover, in some real-world applications, due to data security concerns, the labeled training data is no longer accessible for further optimization once a model has been trained. To overcome this challenge, the most straightforward solution is to collect and annotate new data from the target scenario, but this approach requires considerable resources.

In recent years, researchers have explored source-free adaptation to address the above issues. Numerous SFA methods have been proposed that can be broadly classified into three categories: self-training, domain-invariant features, and adaptive normalization. Self-training (Li et al., 2021; Chen et al., 2023) aims to select high-quality pseudo-labels or samples to fine-tune the detector. Domain-invariant features (Chu et al., 2023; Huang et al., 2021) explore the strategy of mitigating discrepancies in feature distributions between cross domains by employing techniques such as adversarial learning (Chu et al., 2023) or contrastive learning (Huang et al., 2021). Adaptive normalization (Klingner et al., 2022; Hao et al., 2024) seeks to investigate the statistical method of BatchNorm parameters from the source domain to the target domain. Although these methods can improve detector performance in the target domain, they do not consider the potential decrease in performance in the source domain after SFA training, as shown in Fig. 1 (a). This issue arises primarily for two reasons: (1) the large semantic gap between the source and target domains often leads to forgetting or interference

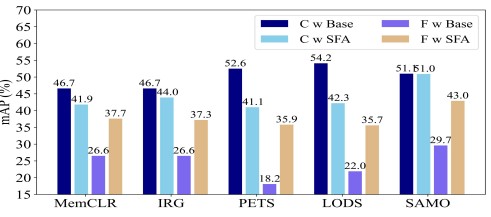
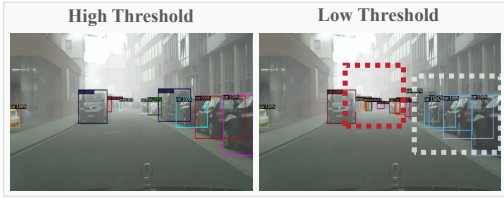

(a) **Comparisons with various SFA methods**  (b) **Performance on different semantic regions**

Figure 1: (a) reports the performance of the base Faster R-CNN (trained solely on the source domain) and SFA-trained variants on both source and target domains from C→F adaptation, denoted as "C w Base", "C w SFA", "F w Base" and "F w SFA". (b) shows visual results across different semantic regions of the target-domain image, under high and low NMS confidence thresholds.

with source knowledge during target domain fine-tuning; (2) Noise in pseudo-labels from the target domain, especially those with low confidence, can negatively impact the detector performance. Fig. 1(b) shows noise results in the pseudo-labels under different confidence levels.

To address the above issues, we propose a novel Semantic-Aware Model Optimization (SAMO) framework. As shown in Fig. 2, the core of this framework is to explore the relationship between semantic variations in the target domain images and model optimization. In target-domain images, the distribution of source knowledge, target knowledge, and label noise varies significantly across different confidence spectrum. High-confidence regions contain richer source knowledge and less noise, enabling model optimization to enhance target-domain performance with minimal degradation on the source domain. In contrast, low-confidence regions involve less source knowledge and more noise, yielding substantial improvements on the target domain but causing a larger performance drop on the source domain. To address this trade-off, we propose the SAMO framework, which integrates the complementary advantages of different confidence spectrum to achieve a balanced performance across both domains.

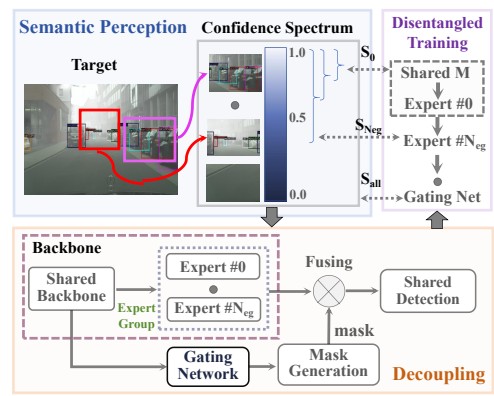

Figure 2: The structure of the core idea of our SAMO. "Shared M" represents the shared modules including the shared backbone and detection, and $S$ denotes a set of confidence intervals.

Specifically, we first perform semantic perception to partition target-domain regions into a confidence spectrum (from high to low) based on pseudo-labels generated by a static base detector. Building on this, we propose an adaptive optimization strategy that guides detector learning across different spectrum ranges via detector decomposition and disentangled training. The detector is divided into three components: shared modules, an expert group, and a gating network. During disentangled training, each module of the disentangled detector is assigned a spectrum range according to its confidence level and employs a tailored strategy, where its loss is computed based on the corresponding confidence range. Meanwhile, these modules are progressively optimized following the sequence illustrated in the "Disentangled Training" flowchart in Fig. 2. The shared modules are primarily fine-tuned on high-confidence regions, enabling adaptation to target-domain semantics while minimizing catastrophic forgetting and cross-domain interference. Each expert specializes in modeling semantics within its assigned confidence range, and the gating network identifies the semantic type of each region to integrate knowledge from different experts. Together, this enables adaptive object detection in the target domain. By applying our framework, the detection model effectively mitigates performance drift in source-free adaptive object detection.

In summary, our main contributions are as follows.

- We propose a novel semantic-aware model optimization framework to encode the relationship between semantic awareness and model optimization, addressing the issue of performance drift in SFA object detection.

- We decouple the detector and design a gating network based on semantic perception, enabling effective learning from diverse target-domain semantics and adaptive application of the acquired knowledge.

- We introduce a disentangled training strategy to progressively train each part of the detector, effectively mitigating the forgetting of source domain knowledge.

- Our method demonstrates effectiveness across various domain adaptation scenarios.

## 2 RELATED WORK

**Unsupervised Domain Adaptive Detection** Unsupervised Domain Adaptive (UDA) Detection aims to improve the performance of detectors on target domain by using labeled source domain images and unlabeled target domain images. Currently, numerous methods have been proposed, which can be broadly categorized into: domain-invariant feature, self-training, image augmentation, and text-driven approaches. Domain-invariant features primarily explore strategies such as adversarial learning (Zhou et al., 2022; Zhao & Wang, 2022), contrastive learning (Cao et al., 2023) and consistency learning (Zhou et al., 2023) with perturbations to reduce the discrepancy in features or feature distributions between the source and target domains. Self-training methods (Chen et al., 2022; Deng et al., 2023) focus on predicting high-quality samples or pseudo-labels, then improving the adaptability of detectors on the target domain by using them. Image augmentation (Yu et al., 2022; Arruda et al., 2022) leverages image style transfer techniques to transform source domain images into the images with target-domain style. Recently, text-driven domain adaptation methods (Li et al., 2023; He et al., 2023) have been proposed, which utilize the zero-shot capabilities of text model (*i.e.*, CLIP (Radford et al., 2021) and GLIP (Li et al., 2022a)) to enhance the generalization ability of detectors. While these methods can effectively improve the detector's performance on the target domain for UDA detection, they still rely on the source domain during adaptive training. As a result, their application in source-free domain adaptation is limited.

**Source-free Adaptive Detection** Source-free Adaptive (SFA) Detection is to solve the problem of UDA Detection when the source domain is inaccessible. To end this, researchers have conducted extensive studies. The work (Chen et al., 2023) considers pseudo-label generation for target domain images from both high-confidence and low-confidence perspectives. The work (Zhang et al., 2023) obtains high-quality pseudo-labels by identifying an appropriate threshold for each category. Different from UDA detection, this approach in SFA detection needs to consider accumulated error from noise in pseudo-labels due to the lack of labeled data. The work (VS et al., 2023a) employs graph-based strategy to learn robust feature representations. The work (Huang et al., 2021) enhances the feature representation of foreground objects using memory-based contrastive learning. Although these methods effectively improve the detector's performance on the target domain, they overlook the negative impact of semantic differences between the source and target domains, such as normal weather versus foggy weather, on the learned source domain knowledge. In this work, we propose a new semantic-aware model optimization framework, which reduces the performance drift in domain adaptation by establishing a relationship between semantic awareness and model optimization.

**Teacher-Student Model** The teacher-student framework employs a teacher network to guide the optimization of the student network. In SFA detection, this framework has been widely adopted in numerous prior studies. The work of (Liu et al., 2023) introduces dual dynamic-static teachers, where the dynamic teacher is updated by dynamically exchanging weights between the student and the static teacher to reduce error accumulation. The method of (Li et al., 2022b) utilizes style discrepancy between original and augmented images in the target domain for self-supervised learning to alleviate domain shift. The work in (Yoon et al., 2024) addresses the challenges of pseudo-label prediction in target images using a teacher-student framework, with a focus on both high- and low-confidence regions. In this work, we construct the SAMO framework by incorporating expert groups and a gated networks into the teacher-student model.

**Mixture-of-Experts** The Mixture-of-Experts (MoE) consists of multiple units, each with a sub-network specialized for a specific task. Recently, MoE has shown strong performance across various vision tasks. Huang et al. (Huang et al., 2025) combine residual learning with MoE, highlighting limitations of policy formulation on diverse object datasets. Lee et al. (Lee et al., 2024) propose a Mixture of Domain Experts framework, incorporating domain-adaptive routing and collaborative

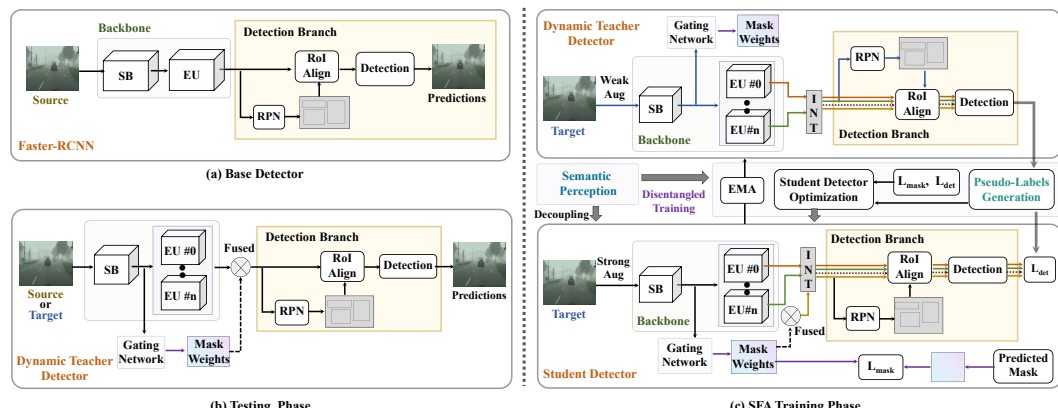

Figure 3: Architecture of our SAMO. "EU" denotes the expert unit module, "SB" represents the shared backbone modules, "#0" represents the index 0 of the expert unit in the expert group.

loss to tackle CTTA challenges. Jain et al. (Jain et al., 2023) introduce a dataset-aware MoE, where experts specialize in specific datasets and each dataset token is routed to its corresponding expert.

**Differences from these methods.** In this work, we propose a novel SAMO approach that leverages semantic awareness to guide model optimization for SFA object detection. The key innovation lies in partitioning target-domain semantics into a confidence spectrum, explicitly analyzing the distribution of source- and target-domain knowledge across the spectrum. Based on this, we design an expert group and employ a disentangled training strategy, enabling each detector module to learn semantics corresponding to specific confidence ranges, further balancing source and target domain knowledge. During testing, the gating network dynamically recognizes regional semantics, facilitating adaptive detection. Overall, SAMO effectively mitigates performance drift in SFA detection.

## 3 PROPOSED METHOD

### 3.1 OVERALL FRAMEWORK

Fig. 3 illustrates our SAMO framework for SFA object detection. It comprises a base detector (Fig. 3(a)), dynamic teacher and student detectors. Compared with the base detector, the teacher and student models incorporate an additional expert group and gating network. Following (VS et al., 2023a), we adopt Faster R-CNN as the base detector, which is decomposed into a backbone and a detection branch (Fig. 3(a)).

As shown in Fig. 3(c), during SFA training, we begin with a perceptual analysis of target-domain images to derive a confidence spectrum. Guided by this spectrum, the student detector is decoupled into the shared modules, expert group, and gating module. In the disentangled training, each module is optimized within its assigned confidence interval, using pseudo-labels and gating masks generated accordingly. Following the sequence in Fig. 2, we progressively train all modules of the student detector, and finally update the teacher detector via exponential moving average (EMA). In addition, during testing phase, given a test image, we first extract a feature map using the shared backbone. The expert group then produces expert-specific feature maps, while the gating network generates a mask to adaptively fuse them into a unified feature map. Finally, fused feature map passed to the detection head for object detection. Fig. 3(b) presents the process of the testing phase.

### 3.2 SEMANTIC PERCEPTION

As shown in Fig. 2, target-domain images exhibit significant semantic variations, leading to varying confidence scores from a base detector trained on the source domain. Moreover, different semantics differ in the amount of source-domain knowledge they retain and the level of noise in pseudo-labels they generate. These factors are overlooked by previous SFA-OD methods. In this work, we leverage semantic perception to design effective learning strategies tailored to different semantics and

enable the detector to adaptively switch between semantic-specific knowledge during testing, thereby achieving balanced performance across both source and target domains.

Specifically, we investigate the semantic perception of the detector for image regions in both SFA training and testing. In SFA training, target-domain regions are partitioned into a confidence spectrum based on pseudo-labels from a static base detector: high-confidence regions preserve richer source-domain knowledge with less noise, while low-confidence regions retain less source-domain knowledge with higher noise. The disentangled training strategy then assigns each detector module to a specific spectrum range, using thresholded pseudo-labels for supervision. At test time, a gating network identifies the semantic type of each region in the input images.

### 3.3 DECOMPOSITION OF THE DETECTOR

Following semantic perception, we decompose the base detector by integrating an expert group module and a gating network. These submodules are optimized using a disentangled training strategy, enabling fine-tuning across different semantic types.

**Shared Modules** The shared module (SM) is divided to retain more source domain knowledge while enabling effective fine-tuning on the target domain. Additionally, it serves to constrain the size of the expert module by mitigating the parameter overhead introduced by the expert group. As shown in Fig.3, the shared module comprises the shared backbone module $f_{bs}$ and the detection branch $f_{det}$. Consider the entire backbone network consisting of $L$ layers. The shared backbone module $f_{bs}$ comprises the first $L_0$ layers ($L_0 < L$). The detection branch is composed of the Region Proposal Network (RPN) and the RCNN module.

**Expert Group** The expert group is designed to balance learning diverse target-domain semantics with preserving source-domain knowledge. As shown in Fig. 3, it consists of $N_{eg}$ expert units, where each expert unit $f_{be}^i$ comprises the layers from $L_0$ to $L$ in the detector's backbone. In addition, an adaptive fusion mechanism is employed, where the outputs of the expert units are integrated based on a gating mask $M$ produced by the proposed gating network. Let $F_i^{eg}$ denote the output feature map of the $i$-th expert unit, and $F_m$ the fused feature map, which is obtained as follows:

$$\mathcal{F}_\mathrm{m} = \sum_{i=0}^{N_{eg}-1} F_i^{eg} \cdot M_i \tag{1}$$

**Gating Network** To achieve regional-level semantic perception and accurately assess the performance of expert units on input image regions, we developed a gating network $f_{gate}$. As shown in Fig.4, $f_{gate}$ adopts a compact encoder–decoder architecture with three down-sampling blocks, two up-sampling blocks, and a final classification layer, where each sampling block is followed by (Ronneberger et al., 2015).

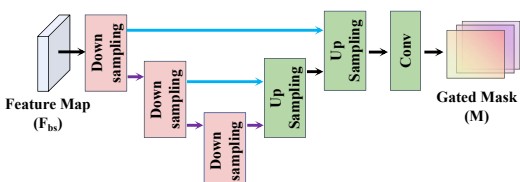

Figure 4: The structure of the gating network.

As illustrated in Fig.3, the shared module $f_{bs}$ extracts feature maps $F_{bs}$, from which the gating network produces a probability map $P \in \mathbb{R}^{N_{eg} \times H \times W}$, where $N_{eg}$ is the number of expert units. The final gating mask $M$ is then derived from $P$.

$$\mathcal{O}_{i,j} = \begin{cases} 1.0, P_i - P_j > e \\ 0.0, otherwise \end{cases} \quad , \qquad \mathcal{M}_i = \prod_{j=-1}^{i-1} O_{i,j} \cdot (1 - \sum_{k=i+1}^{N_{eg}-1} M_k) \tag{2}$$

where $i$ and $j$ are the channel indices of the probability map $P$, and $e$ are a constant threshold. $O_{0,-1}$ is a matrix with all elements equal to 0. $O_{i,j}$ measures the performance difference between the expert units $i$ and $j$. If the prediction probability of $i$-th expert unit exceeds that of $j$-th expert unit by more than a threshold $\epsilon$, we set $O_{i,j} = 1$, indicating that $i$-th expert unit outperforms $j$-th expert unit. During disentangled training, experts 0 to $N_{eg} - 1$ are optimized over progressively broader confidence spectra, leading to reduced source-domain knowledge and greater sensitivity to pseudo-label noise. To evaluate the overall performance of all expert units, we adopt a progressive strategy: for the $i$-th expert, regions where later experts ($i+1$ to $N_{eg}-1$) perform better are excluded, while regions outperforming all earlier experts (0 to $i-1$) are retained for its final evaluation.

## 3.4 DISENTANGLED TRAINING STRATEGY

We further propose a disentangled training strategy, in which a set of confidence spectra obtained from semantic perception is progressively used to optimize the detector's decoupled modules. We also present the detector's inference pipeline during testing.

**Training Pipeline.** As shown in Fig. 2, we progressively optimize the shared modules, expert units, and gating module along a descending confidence spectrum, balancing source-domain knowledge preservation with target-domain adaptation. Formally, we define a set of confidence intervals $\mathcal{S} = \{(\beta_i^{dw}, \beta_i^{up}] \mid i \in [0, N_{eg})\}$, where intervals with larger $i$ values include lower-confidence samples. We first train the shared modules and expert unit of 0-th, using the highest-confidence interval $(\beta_0^{dw}, \beta_0^{up}]$. Within this range, NMS thresholds are applied to select reliable samples, which are optimized using detection losses to maintain source-domain features while initiating target-domain learning. We then progressively include lower-confidence intervals $(\beta_i^{dw}, \beta_i^{up}]$ to fine-tune the expert unit of 1-th through $(N_{eg} - 1)$-th. Finally, using the full set $\mathcal{S}$, we generate region masks and train the gating module with a segmentation loss.

**Pseudo-label Generation.** In the training pipeline, both gating mask and pseudo-labels must be generated. The gating mask is used to guide the training of the gating network, while the pseudo-labels are used to fine-tune the shared modules and expert group module. Given a target-domain image $x_t$, the base detector outputs predictions $D_{\mathrm{pro}} = \{(p_j, l_j, b_j) \mid j \in [0, N_{\mathrm{pro}}]\}$, and the dynamic teacher detector produces $D_{\mathrm{dt}} = \{(p_j, l_j, b_j) \mid j \in [0, N_{\mathrm{dt}}]\}$, where $N_{\mathrm{pro}}$ and $N_{\mathrm{dt}}$ denote the number of proposals, and $p_j$, $l_j$, and $b_j$ represent the confidence score, class label, and bounding box of the $j$-th proposal, respectively. For $i$-th exprt unit, proposals from $D_{\mathrm{pro}}$ with probabilities within $[\beta_i^{\mathrm{dw}}, \beta_i^{\mathrm{up}})$ are selected, and their corresponding image regions are assigned a value of 1, while all others are set to 0, forming an intermediate binary mask $\xi_i$. The final gating mask $\hat{m} \in \mathbb{R}^{N_{eg} \times H \times W}$ is then obtained by aggregating these class-wise masks.

$$\hat{m}_i = \xi_i \cdot (1 - \sum\nolimits_{k=0}^{i-1} \xi_k) \tag{3}$$

where $\hat{m}_i$ is obtained by removing the mask regions of all previous $(t-1)$ expert units from $\xi_i$. As there is no $(-1)$-th expert unit, we set $\xi_{-1}$ as a zero matrix. We then generate pseudo-labels $\hat{y}_i$ for each confidence interval by applying Non-Maximum Suppression (NMS) with a fixed confidence threshold $\delta$ and IoU threshold $\epsilon$.

$$\hat{y}_i = NMS(D_{dt}, \delta_i, \epsilon_i) \tag{4}$$

where the confidence threshold $\delta_i$ and IoU threshold $\epsilon_i$ are associated with the $i$-th confidence interval, respectively. Notably, compared to the dynamic teacher and student detectors, the base detector, trained solely on the source domain, produces more stable predictions on target domain images. Therefore, we utilize its predictions to generate semantic partitions and corresponding gating masks (*i.e.*, $\hat{m}$) for the target domain.

**Student Detector Optimization.** As outlined in the training pipeline, we progressively optimize the shared modules, expert units, and the gating network. This training strategy enables each module to focus on learning the corresponding semantic knowledge from the target domain, thus reducing internal interference and negative transfer effects. The loss functions for the shared and expert modules can be expressed in the following unified form.

$$\mathcal{L}_i^{op} = \mathcal{L}_{det}(x^t, \hat{y}_i^t, f_i, \theta_i) \tag{5}$$

where $L_{det}$ is detection loss used in Faster-RCNN (Ren et al., 2017), and $f_i \in \{(f_{sm}, f_{be}^0), .., f_{be}^{N_{eg}}\}$. $f_{sm}$ represents the shared module, which includes $f_{bs}$ and $f_{det}$. $\theta_i$ denotes the network parameters of $f_i$, and the loss function $\mathcal{L}_i^{op}$ is solely responsible for optimizing the parameters of $\theta_i$. After training the shared modules and expert units of the student detector, we optimize the gating module using the output feature map $F_{bs}$ from the shared backbone $f_{bs}$ and the gating mask $\hat{m}$. Moreover, we apply the Dice loss(Milletari et al., 2016) to guide this training.

$$\mathcal{L}_{mask} = L_{Dice}(f_{gate}(F_{bs}), \hat{m}) \tag{6}$$

**Teacher Detector Update.** Let $\theta_t$ and $\theta_s$ denote the parameters of the dynamic teacher and student detectors, respectively. Following (Yoon et al., 2024), we update the teacher parameters via EMA after each iteration.

$$\theta_t = \alpha \cdot \theta_t + (1 - \alpha) \cdot \theta_s \tag{7}$$

Table 1: Experiments from C→F adaptation using AP (%).

| Method | BB | C | F |
|---|---|---|---|
| Source Only | V16 | 52.6 | 18.2 |
| PETS* (Liu et al., 2023) | V16 | 41.1 | 35.9 |
| Source Only* | V16 | 54.2 | 22.0 |
| LODS* (Li et al., 2022b) | V16 | 42.3 | 35.7 |
| Source Only | R50 | 46.7 | 26.6 |
| IRG (VS et al., 2023a) | R50 | 44.0 | 37.3 |
| MemCLR (VS et al., 2023b) | R50 | 41.9 | 37.7 |
| LPLD* (Yoon et al., 2024) | R50 | 40.9 | 38.1 |
| Source Only ( Our ) | R50 | 51.1 | 29.7 |
| SAMO ( w Base ) | R50 | 47.8 | 42.6 |
| SAMO (w EU0) | R50 | 51.0 | 40.3 |
| SAMO (w EU1) | R50 | 48.7 | 43.9 |
| SAMO | R50 | 51.0 | 43.0 |

Table 2: Experiments from M→C adaptation using AP (%).

| Method | BB | M | C |
|---|---|---|---|
| Source Only* | V16 | 69.7 | 41.4 |
| PETS* (Liu et al., 2023) | V16 | 63.8 | 48.5 |
| Source Only* | V16 | 64.0 | 33.9 |
| LODS* (Li et al., 2022b) | V16 | 59.2 | 41.0 |
| Source Only | R50 | 67.8 | 33.3 |
| MemCLR* (VS et al., 2023b) | R50 | 64.9 | 46.8 |
| IRG* (VS et al., 2023a) | R50 | 64.8 | 47.5 |
| LPLD* (Yoon et al., 2024) | R50 | 58.7 | 50.0 |
| Source Only ( Our ) | R50 | 69.1 | 44.1 |
| SAMO ( w Base ) | R50 | 63.8 | 53.9 |
| SAMO (w EU0) | R50 | 69.1 | 48.8 |
| SAMO (w EU1) | R50 | 63.5 | 53.7 |
| SAMO | R50 | 67.0 | 51.9 |

### 3.5 OVERALL OPTIMIZATION

The training process of the proposed SAMO framework can be divided into two stages. In the first stage ($S1$), we train a base detector using object detection loss in Faster-RCNN on the source domain. In the second stage ($S2$), we perform a domain-adaptive optimization with disentangled training strategy (see Sec .3.4 ) on the target domain.

## 4 EXPERIMENTS

### 4.1 DATASETS

We perform the evaluation in the following adaptation scenarios. Following (Li et al., 2021), we use the standard mean average precision (mAP) at an IoU threshold of $0.5$ as our evaluation metric for all experiments.

**C→F adaptation.** Cityscapes (Cordts et al., 2016) (**C**) is an outdoor street scene dataset, consisting of $2,975$ training images and $500$ test images. FoggyCityscapes (Sakaridis et al., 2018) (**F**) is synthesized from the Cityscapes, and contains three levels of fog intensity (*i.e.*, 0.1, 0.05, 0.02). In this experiment, we use Cityscapes as the source domain and FoggyCityscapes with the highest fog intensity of $0.02$ as the target domain. The FoggyCityscapes validation set is used for evaluation.

**M→C adaptation.** Sim10k (Johnson-Roberson et al., 2017) (**M**) is a synthetic dataset from GTA5 with 10,000 labeled images, where 8,550 images are randomly sampled for training and the rest for testing. In our experiments, Sim10k serves as the source domain and Cityscapes as the target domain (M→C). The Cityscapes validation set is used for evaluation on the target domain, focusing only on the "car" category as in (VS et al., 2023b).

**V→W adaptation.** Pascal VOC (Everingham et al., 2010), a widely used object detection dataset, uses the VOC2007 and VOC2012 training sets for training and the VOC2012 validation set for testing. The Watercolor dataset (Inoue et al., 2018) contains 1,000 training and 1,000 test images. In our experiments, Pascal VOC serves as the source domain and Watercolor as the target domain (V→W), with evaluation conducted on the VOC2012 validation set and the Watercolor test set.

### 4.2 IMPLEMENTATION

Our SAMO framework is implemented on Pytorch. Following (Li et al., 2021), we utilize Faster-RCNN (Ren et al., 2017) with VGG16 (Liu & Deng, 2015) or ResNet50 (He et al., 2016) pre-trained on ImageNet (Krizhevsky et al., 2012) as the base detector. $N_{eg}$ in the expert group is set to 2. $\beta_0^{dw}$, $\beta_0^{up}$, $\beta_1^{dw}$, and $\beta_1^{up}$ in the confidence intervals are set to 0.95, 1.1, 0.01, and 0.95, respectively. In Eq. 4, $(\delta_{f_{sm}}, \epsilon_{f_{sm}})$ and $(\delta_0, \epsilon_0)$ both are set to $(0.9, 0.6)$. $\delta_1$ and $\epsilon_1$ are set to 0.8 and 0.6. $\alpha$ is set to 0.999 in Eq. 7. $e$ in Eq.2 is set to 0.6. Besides, the optimizer employs the Adam with a momentum of 0.9 and weight decay of $1e-4$. The learning rate is set to $1e-4$.

Table 3: Experiments from V→W adaptation using AP (%).

| Method | BB | V | W |
|---|---|---|---|
| Source Only* | V16 | 62.4 | 29.8 |
| PETS* (Liu et al., 2023) | V16 | 54.8 | 36.0 |
| Source Only* | V16 | 74.0 | 30.9 |
| LODS* (Li et al., 2022b) | V16 | 71.5 | 33.0 |
| Source Only* | R50 | 88.2 | 45.1 |
| MemCLR* (VS et al., 2023b) | R50 | 84.5 | 51.3 |
| IRG* (VS et al., 2023a) | R50 | 85.0 | 52.1 |
| LPLD* (Yoon et al., 2024) | R50 | 79.7 | 53.5 |
| Source Only ( Our ) | R50 | 89.8 | 44.4 |
| SAMO ( w Base ) | R50 | 83.1 | 51.6 |
| SAMO (w EU0) | R50 | 89.1 | 47.0 |
| SAMO (w EU1) | R50 | 82.6 | 53.6 |
| SAMO | R50 | 87.9 | 52.7 |

Table 4: Analysis with different backbones from two adaptations (*i.e.*, C→F and M→C) using AP (%).

| Method | BB | C→F | | M→C | |
|---|---|---|---|---|---|
| | | C | F | M | C |
| Source Only ( Our ) | V16 | 43.9 | 28.4 | 66.0 | 40.6 |
| SAMO ( w Base ) | V16 | 42.1 | 37.3 | 60.6 | 48.5 |
| SAMO (w EU0) | V16 | 45.2 | 36.1 | 66.1 | 41.4 |
| SAMO (w EU1) | V16 | 43.3 | 38.2 | 63.4 | 48.7 |
| SAMO | V16 | 44.4 | 38.3 | 65.9 | 49.4 |
| Source Only ( Our ) | R50 | 51.1 | 29.7 | 69.1 | 44.1 |
| SAMO ( w Base ) | R50 | 47.8 | 42.6 | 63.8 | 53.9 |
| SAMO (w EU0) | R50 | 51.0 | 40.3 | 69.1 | 48.8 |
| SAMO (w EU1) | R50 | 48.7 | 43.9 | 63.5 | 53.7 |
| SAMO | R50 | 51.0 | 43.0 | 67.0 | 51.9 |

## 4.3 COMPARISON WITH OTHER METHODS

In this section, we compare our approach against other SFA methods with publicly available implementations, including MemCLR (VS et al., 2023b), IRG (VS et al., 2023a), LPLD (Yoon et al., 2024), LODS (Li et al., 2022b), and PETS (Liu et al., 2023). Methods without reproducible code (e.g., SF-UT (Hao et al., 2024) and LPU (Chen et al., 2023)) are excluded for fairness. "Source Only" refers to testing the source-trained detector without adaptation. "SAMO (w Base)" denotes a traditional teacher–student framework, while "SAMO (w EU0)" applies our SAMO framework using only the 0-th expert unit. "SAMO" reports results from the full SAMO framework with all expert units and the gating network. "*" denotes our reproduced results due to unavailable training weights.

**C→F adaptation.** As shown in Tab. 1, compared to "Source only", although MemCLR (VS et al., 2023b) and IRG (VS et al., 2023a) improve the detection performance on the unseen target domain, their performance on the source domain drops significantly. In contrast, our approach demonstrates a clear advantage in overall performance. As observed in Tab. 1, in our SAMO framework, the first expert unit enhances the performance of the target domain while maintaining high mAP score in the source domain. The second expert unit further improves the performance of the target domain, although at the cost of reduced accuracy in the source domain. By introducing a gated network, our method achieves mAP of $43.0\%$ on the target domain and mAP of $51.0\%$ on the source domain, demonstrating strong detection performance across both domains.

**M→C adaptation.** Tab. 2 reports the results from the M→C adaptation. Similarly, IRG (VS et al., 2023a) and LPLD (Yoon et al., 2024) effectively improve detection performance on the target domain, but at the cost of a performance drop on the source domain. In contrast, our method achieves high detection performance on both the source domain with mAP of $67.0\%$ and the target domain with mAP of $51.9\%$. These results further validate the effectiveness of our approach in reducing performance drift for SFA object detection.

**V→W adaptation.** Tab. 3 shows the evaluation results for the V→W adaptation. As shown in Tab. 3, compared to other existing SFA detection methods, our approach achieves a detection performance of $52.7\%$ on the target domain while also maintaining a performance of $87.9\%$ on the source domain. Moreover, in comparison with "Source Only ( Our )", our method not only delivers an improvement of $8.3\%$ on the target domain but also mitigates the performance drop on the source domain. These results clearly demonstrate the effectiveness of our approach in reducing performance drift.

To this end, as evidenced by the experimental results across Tab. 1 to Tab. 3, the proposed SAMO framework effectively enhances detection performance on unseen target domains while preserving source-domain accuracy.

## 4.4 ABLATION STUDY

In this section, we analyze the effects of different backbones, the number of expert units, and their parameter sizes from the C→F adaptation. Additional ablations are provided in the **Appendix**.

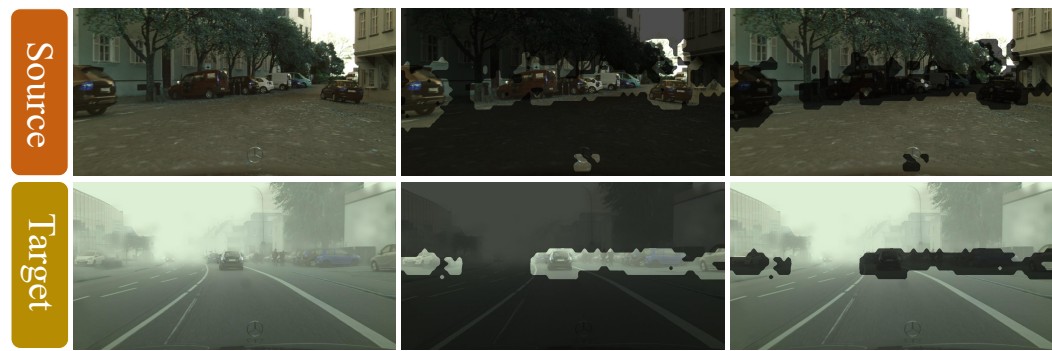

Figure 5: Visualization of the gated mask from C→F. The first row is the results for the source domain, while the second row shows those for the target domain. The first column displays the original images. The last two columns illustrate the heatmaps generated from $M_0$ and $M_1$, respectively. Dark regions represent mask values of 0, and the remaining regions correspond to mask values of 1.

Table 5: Analysis of the expert unit number.

| Method | Backbone | C | F |
|---|---|---|---|
| Source only | ResNet50 | 51.1 | 29.7 |
| $N_{eg} = 1$ | ResNet50 | 47.8 | 42.6 |
| $N_{eg} = 2$ | ResNet50 | 51.0 | 43.0 |
| $N_{eg} = 3$ | ResNet50 | 50.6 | 43.9 |

Table 6: Analysis with different parameter number of the expert unit.

| mAP | Source only | Base | L3 | L2+L3 | L1+L2+L3 |
|---|---|---|---|---|---|
| C | 51.1 | 47.8 | 50.4 | 51.0 | 50.1 |
| F | 29.7 | 42.6 | 41.7 | 43.0 | 44.0 |
| P (M) | – | – | 54.0 | 63.2 | 64.9 |

**Analysis with Different Backbones.** In Tab. 4, we further analyze the effectiveness of different backbones (*i.e.*, VGG16 and ResNet50) from C→F and M→C adaptations on Faster-RCNN detector. As shown in Tab. 4, our method improves the detector's performance on the target domain while maintaining high performance on the source domain across both VGG16 and ResNet backbones, demonstrating its versatility.

**Analysis of the Expert Unit Number.** In Tab. 5, we analyze the impact of the expert unit number from C→F. As shown in Tab. 5, $N_{eg} = 2$ yields mAP scores of 51.0% and 43.0% on the source and target domains, surpassing $N_{eg} = 1$. While $N_{eg} = 3$ achieves the highest target-domain mAP, its slightly lower source score compared to $N_{eg} = 2$ may stem from the confidence interval assignment and threshold $e$ in Eq.2, which limit source-knowledge utilization. Overall, increasing $N_{eg}$ enhances detector performance and achieves a favorable trade-off between source and target domains.

**Analysis of Expert Unit Parameters.** In Tab. 6, we analyze the effect of varying the parameter size of the expert unit. The ResNet50 backbone in Faster R-CNN consists of "stem", "layer1", "layer2", and "layer3" (denoted as $L0$–$L3$). "P (M)" indicates the total learnable parameters of the expert group, measured in MB. We test different expert unit configurations using these modules. As shown in Tab. 6, larger expert units amplify the impact of the expert group and gating mechanism, and our method also demonstrates a more significant reduction in performance drift.

## 5  CONCLUSION

In this paper, we propose a novel semantic-aware model optimization framework to address performance degradation in SFA detection. Specifically, we first categorize region semantics in the target domain into a spectrum of confidence levels (ranging from high to low), enabling semantic perception. Based on this, we then structurally decouple the base detector and employ a disentangled training strategy, allowing different modules to focus on a distinct portion of the semantic spectrum. Finally, we introduce a gating network module that recognizes semantic types and adaptively integrates the learned knowledge. Extensive experiments demonstrate that our method significantly mitigates the performance degradation issue in SFA object detection.

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

# 6 APPENDIX

## 6.1 HYPERPARAMETER ANALYSIS.

In Tab. 7, we analyze the effect of the threshold parameter $e$ in Eq. 2 on the adaptive fusion performance of the gating network. The experiment is conducted on ResNet-50 from C→F adaptation. "SAMO (w/ EU0)" and "SAMO (w/ EU1)" denote applying our SAMO framework with only the 0-th or 1-th expert unit, respectively. "SAMO" reports results under varying values of $e$ in Eq. 2.

As shown in Tab. 7, when $e = 0.6$, the detector achieves strong performance on both the source and target domains. However, a larger threshold $e$ assigns more image regions to the 0-th expert unit. This increases reliance on source-domain knowledge while reducing the use of target-domain knowledge. As a result, source-domain performance improves, but target-domain performance gradually declines. Conversely, an excessively small $e$ assigns more regions to the 1-th expert unit. This emphasizes target-domain knowledge while underutilizing source-domain knowledge, leading to higher performance on the target domain but lower performance on the source domain.

Table 7: Analysis with the threshold parameter $e$.

| mAP | SAMO (w EU0) | SAMO (w EU1) | SAMO | | | | |
|---|---|---|---|---|---|---|---|
| | | | 0.0 | 0.2 | 0.4 | 0.6 | 0.8 |
| C | 51.0 | 48.7 | 49.6 | 50.2 | 50.5 | 51.0 | 51.1 |
| F | 40.3 | 43.9 | 43.5 | 43.7 | 43.4 | 43.0 | 40.5 |

## 6.2 VISUALIZATION OF GATING MASKS

In Fig. 6 and 7, we present more visual results of the gating masks from C→F and M→C adaptations. As shown in Fig. 6 and 7, our gating network and the mask generation strategy (Eq. 2) effectively perceive the semantics of input image regions and adaptively switch and apply expert knowledge across different semantic regions.

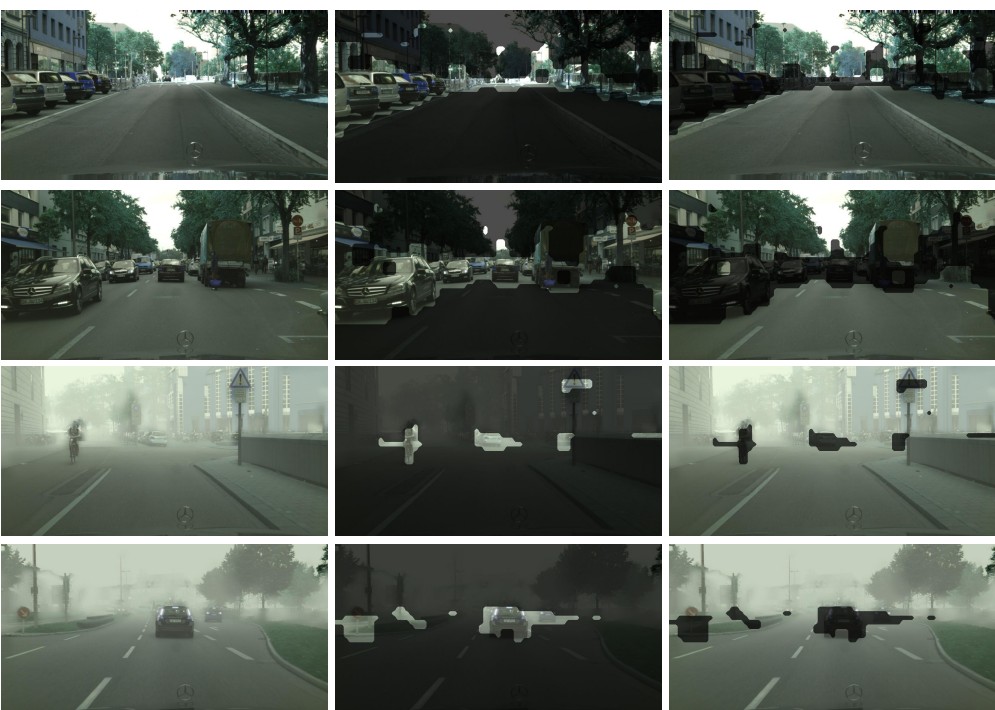

Figure 6: Visualization of gated masks from C→F. The first two rows correspond to the source domain (*i.e.*, C) and the last two to the target domain (*i.e.*, F). The first column shows the original images, while the last two columns present heatmaps from $M_0$ and $M_1$, where dark regions indicate mask value 0 and others indicate 1.

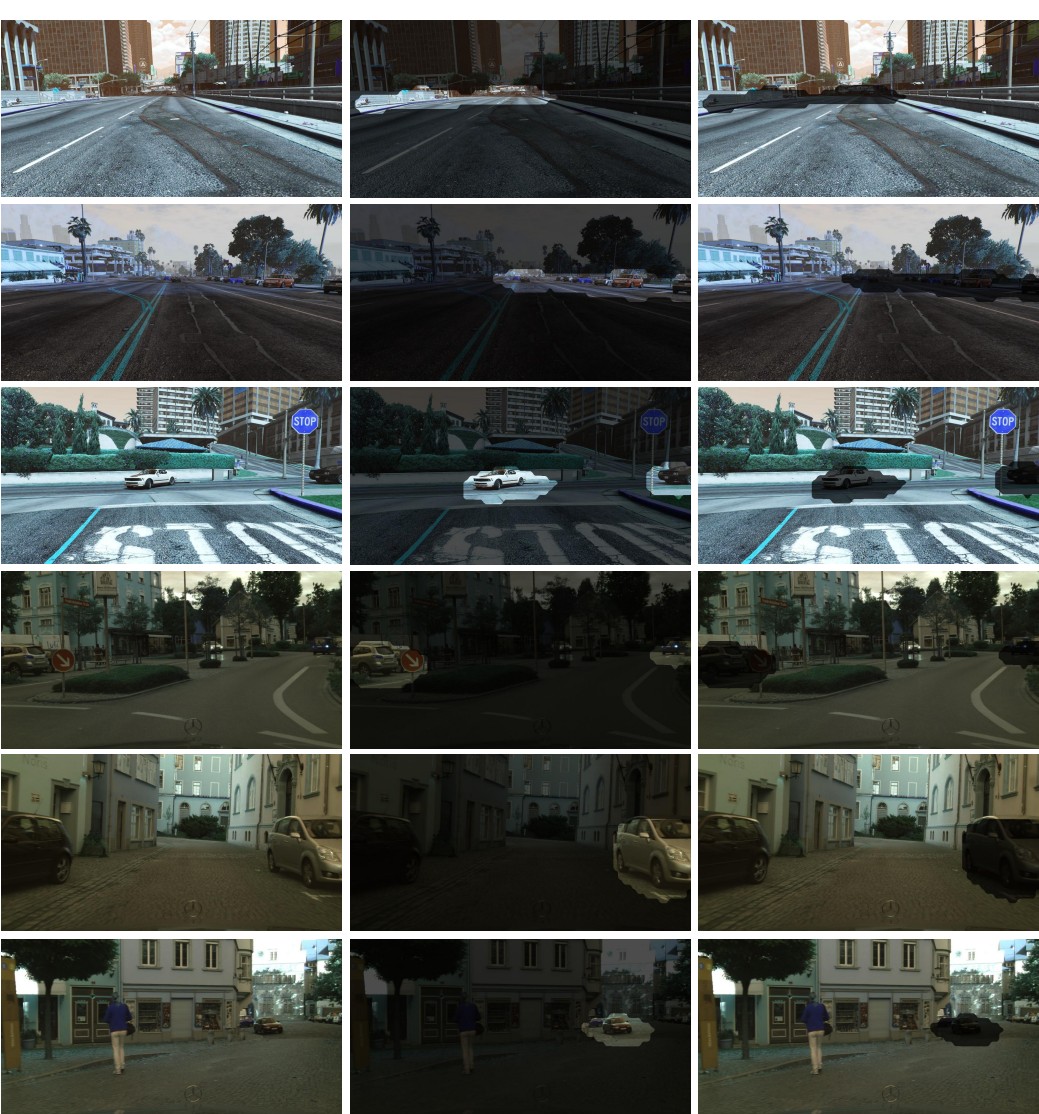

Figure 7: Visualization of the gated mask from M→C. The first three rows correspond to the source domain (*i.e.*, M) and the last three to the target domain (*i.e.*, C). The first column shows the original images, while the last two columns present heatmaps from $M_0$ and $M_1$, where dark regions indicate mask value $0$ and others indicate $1$.

### 6.3 THE USE OF LARGE LANGUAGE MODELS (LLMS)

In this work, we primarily use LLMs to refine the grammar and wording of research papers.

