# OpenReview forum: "SAMO: Semantic-Aware Model Optimization for Source-Free Domain Adaptive Object Detection"
_ICLR.cc/2026/Conference — ICLR 2026 Conference Withdrawn Submission_

### Official Review · Reviewer_7bDa · 2025-10-26

**Soundness:** 3
**Presentation:** 3
**Contribution:** 2
**Rating:** 2
**Confidence:** 4

**Summary:**

This paper studies source-free domain adaptive object detection (DAOD), which aims to adapt a detector pre-trained on a labeled source domain to an unlabeled target domain without access to source data. To address the common issue of performance degradation in source-free DAOD, the authors propose a Semantic-Aware Model Optimization (SAMO) framework. The key idea is to categorize target-domain regions into a confidence spectrum and design a corresponding optimization strategy through detector decomposition and disentangled training. Experiments on three standard DAOD benchmarks demonstrate that SAMO improves generalization on the target domain while maintaining performance on the source domain.

**Strengths:**

1. The paper identifies two major factors contributing to performance degradation in source-free DAOD and provides experimental evidence supporting these observations (e.g., Fig. 1).

2. The proposed SAMO framework is conceptually coherent. It connects semantic perception and model optimization via a confidence spectrum, enabling disentangled training that helps balance adaptation and knowledge retention.

3. The paper is well organized and easy to follow.

4. The paper provides extensive experiments, including comparisons with state-of-the-art methods, ablation studies, and parameter analysis.

**Weaknesses:**

1. The technical contribution is somewhat limited. While SAMO introduces a semantic-aware perspective, its main mechanism can be viewed as a fusion strategy that combines pseudo labels at different confidence levels through a gating mechanism. The novelty over prior teacher-student or mixture-of-experts–based DAOD frameworks appears incremental.

2. The paper discusses performance degradation and forgetting, but no quantitative evidence is presented during training to support these claims. It would be valuable to include performance curves over training iterations to demonstrate that SAMO effectively stabilizes training and mitigates forgetting.

3. The compared methods and adopted architectures are outdated. The experimental setup primarily relies on Faster R-CNN with VGG16 or ResNet-50 backbones. More recent detectors (e.g., ViT-based detectors such as DINO or Deformable DETR variants) should be included. Moreover, comparisons with recent vision-language model (VLM)–based domain adaptation methods would further strengthen the paper.

**Questions:**

Please see the weakness.

---

### Official Review · Reviewer_YC6c · 2025-10-27

**Soundness:** 2
**Presentation:** 1
**Contribution:** 2
**Rating:** 4
**Confidence:** 3

**Summary:**

This paper proposes a Semantic-Aware Model Optimization (SAMO) framework to address the problem of performance drift in Source-Free Domain Adaptive (SFA) object detection. The core of the method lies in using semantic perception to categorize target-domain regions into a confidence spectrum. Based on this, it designs a disentangled training strategy and a gating network to balance performance across the source and target domains. The paper's experiments are extensive , demonstrating the method's effectiveness on multiple benchmarks. Overall, the work has practical value, but the details need further enhancement.

**Strengths:**

1.The paper introduces a semantic-aware model optimization framework, effectively addressing performance drift in SFA object detection by encoding the relationship between semantic awareness and model optimization.

2.The approach of decoupling the detector and designing a gating network based on semantic perception allows for effective learning from diverse target-domain semantics, ensuring adaptive application of acquired knowledge.

3.The paper proposes a disentangled training strategy to progressively train different parts of the detector, which helps mitigate the forgetting of source domain knowledge, demonstrating an efficient training methodology.

4.The method exhibits strong effectiveness across various domain adaptation scenarios, showcasing its versatility and practical applicability in real-world settings.

**Weaknesses:**

1.The SAMO framework introduces several new components, including expert groups, gating networks, and disentangled training strategies, which significantly increase the model's complexity. The document does not adequately discuss computational efficiency, such as training time, memory usage, or inference speed, which may pose bottlenecks for practical deployment.

2.The SAMO framework introduces a large number of additional hyperparameters, but the authors do not provide any analysis on how to systematically determine these values, nor do they offer a comprehensive ablation study.

3.The paper lacks theoretical analysis to support the effectiveness of SAMO,  which lowers the credibility of the method.

4.The structure of the paper is somewhat disorganized, making it difficult to read. Some figures contain errors or are hard to understand.

**Questions:**

1.Why is the gating network structure in Figure 4 designed in this way, and what are its advantages?

2.What do the arrows in Figure 2, pointing from Expert 0 to subsequent experts, represent?

3.In the Testing Phase in Figure 3, should it be the Student Detector instead?

4.How were the hyperparameters for confidence intervals determined in the IMPLEMENTATION section?

---

### Official Review · Reviewer_yp4D · 2025-10-30

**Soundness:** 2
**Presentation:** 2
**Contribution:** 2
**Rating:** 2
**Confidence:** 4

**Summary:**

This paper proposes a framework called SAMO (Semantic Aware Model Optimization), specifically designed for source-free domain adaptive object detection (SFA-OD). SAMO aims to mitigate the performance degradation of SFA-OD by considering semantic variations in the target domain image and utilizing a confidence-based semantic awareness and decoupled training strategy. Experimental results show that SAMO can reduce performance drift and improve object detection accuracy in the target domain.

**Strengths:**

* The paper is easy to follow.
* The key idea of the gated weights and feature fusion is reasonable.

**Weaknesses:**

1. Limited novelty. Overall, the expert unit module acts as a prototype network, while the gated network provides corresponding weights for each prototype. This idea is very common and has been thoroughly studied in many works, such as [1]. In this respect, the proposed method is not particularly innovative.
2. Misuse of new concepts. This paper uses the concept of MoE to refer to the proposed EU network. However, I believe this is a misconception. This is because the proposed EU network performs a weighted average on each output, with all layers participating in the computation, rather than utilizing the route activation parameters as in large-scale pre-trained models.
3. The so-called MoE may introduce significant computational overhead. However, the paper seems to lack relevant discussion, making it impossible to determine whether the performance improvement comes from a better structural design or more parameters.

[1]Bolya D, Zhou C, Xiao F, et al. Yolact: Real-time instance segmentation. CVPR 2019

**Questions:**

Please refer to Weakness.

---

### Official Review · Reviewer_y8MY · 2025-10-31

**Soundness:** 2
**Presentation:** 3
**Contribution:** 2
**Rating:** 4
**Confidence:** 3

**Summary:**

This paper tackles  source-free domain adaptive object detection. The proposed SAMO framework splits the detector into shared modules (to preserve source knowledge), a set of experts trained on different confidence ranges, and a spatial gating network that fuses expert outputs. Training is “disentangled,” starting from high-confidence regions and progressively moving to lower-confidence ones, with a teacher–student setup and EMA. Experiments on Cityscapes→FoggyCityscapes, Sim10k→Cityscapes, and VOC→Watercolor show solid target gains and relatively restrained source drift, plus ablations on the number/size of experts and gating thresholds.

**Strengths:**

1. Results are generally strong across three benchmarks and two backbones.

2. The concept of using confidence as a proxy for domain shift is not new, but the way it's used here to explicitly manage a trade-off through a MoE architecture is novel and well-executed.

**Weaknesses:**

1. From the system level, the novelty is more in the integration than in the components (teacher–student, pseudo-labeling by confidence, MoE with gating).

2. The system is a bit complex. Not clean and neat enough for high impact.

3. the proposed method introduces many hyperparameters.

4. No training/inference computational cost is reported.

**Questions:**

1. Can the authors provide a systematic ablation over the hyperparameters introduced? It is important to verify that the proposed method does not rely on cherry-picked hyperparameters.

---

### Note · Authors · 2025-11-19

I have read and agree with the venue's withdrawal policy on behalf of myself and my co-authors.